# How Does the COVID-19 Pandemic Affect Pediatric Patients with Intussusception Treated by Ultrasound-Guided Hydrostatic Enema Reduction?

**DOI:** 10.3390/jcm11154473

**Published:** 2022-07-31

**Authors:** Min Yang, Ze-Hui Gou, Jun Wang, Ju-Xian Liu, Bo Xiang

**Affiliations:** 1Department of Pediatric Surgery, West China Hospital of Sichuan University, Chengdu 610041, China; hx2014bsym@163.com; 2Department of Ultrasound Medicine, West China Hospital of Sichuan University, Chengdu 610041, China; hellozehuigou@163.com (Z.-H.G.); teodorojw100@gmail.com (J.W.); liujuxian@wchscu.cn (J.-X.L.)

**Keywords:** COVID-19, pediatric, intussusception, ultrasound, enema, reduction

## Abstract

Background: The pandemic of COVID-19 has significantly influenced the epidemiology of intussusception. Nevertheless, the effects of the COVID-19 pandemic on the operation of ultrasound-guided hydrostatic enema reduction (USGHER) for intussusception have been largely unknown. Methods: The data of pediatric patients with intussusception who were treated by USGHER from January to March of 2019 (Control Group), 2020 (Study Group 1), and 2021 (Study Group 2) in a large Chinese medical institution were retrospectively collected and analyzed. Results: We enrolled 246 patients, including 90 cases in Control Group, 70 in Study Group 1, and 86 in Study Group 2 (*p* = 0.042). The time from the onset of symptoms to the hospital visit and the time from the hospital visit to performing the ultrasound in Study Group 1 was significantly longer than that in Control Group and Study Group 2 (*p* = 0.036, *p* = 0.031, respectively). The number of patients with bloody stool and the longest invaginated length of intussusception in Study Group 1 increased significantly compared with patients in the other two groups (*p* = 0.007, *p* = 0.042, respectively). Comparisons of neither the pressure of enema nor the time of duration when performing USGHER present statistical significance among the three groups (*p* = 0.091, *p* = 0.085, respectively). For all enrolled pediatric patients, there was no perforation case involved, and recurrence of intussusception occurred in few cases. Conclusions: Besides the negative impacts on the incidence of intussusception, the COVID-19 pandemic might have led to the diagnostic delay of intussusception and the deterioration of patients’ clinical manifestations, but it did not significantly affect the operation of USGHER and patients’ clinical outcome.

## 1. Introduction

Coronavirus disease 2019 (COVID-19), caused by the severe acute respiratory syndrome coronavirus 2 (SARS-CoV-2), is an acute respiratory disease with symptoms that vary widely, from mild manifestations to potentially fatal respiratory distress symptoms [1]. Since the outbreak of the COVID-19 pandemic on 2 March 2020 [2], this communicable disease has been hindering social development, wreaking financial havoc, burdening healthcare systems, and causing loss of countless lives. According to the official report from the WHO, as of 27 June 2022, there have been 540,923,532 confirmed cases of COVID-19, including 6,325,785 deaths [3]. 

In order to prevent further spread of this epidemic, many countries have carried out a series of compulsory strategies, such as wearing face masks and checking temperatures in public, cancelling large gatherings, mass testing for SARS-CoV-2, closing schools and public places, and implementing social and physical distancing and even lockdowns. Together with the fear of exposure and risk of infection for patients themselves [4], these preventative measures have resulted in a global decline, owing to the number of emergency department (ED) visits and hospitalizations of many diseases [5,6,7].

Intussusception is one of the most common abdominal emergencies and the most common cause of bowel obstruction in infant patients. It occurs commonly between the ages of 4 and 10 months, with an incidence varying from 0.24 to 2.4 per 1000 live births [8,9]. The classic triad of symptoms associated with intussusception, comprising intermittent vomiting, colicky abdominal pain, and bloody stool (i.e., “currant jelly stool”), is noted in less than half of initial clinical encounters [10]. 

So, the prompt and accurate diagnosis of intussusception for infant patients is challenging, and delayed diagnosis may lead to serious complications, such as bowel necrosis and perforation, diffuse peritonitis, septic shock, and even death [10]. The treatment strategies for intussusception vary from prohibition of drinking, fluid infusion, and enema reduction to surgery, according to the clinical features of intussusception and patients’ health conditions [8,9,10]. Recently, ultrasound-guided hydrostatic enema reduction (USGHER) has gained widespread acceptance due to its safer and more effective advantages for the invasive management of intussusception [11,12,13].

The pandemic of COVID-19 has significantly changed the epidemiology of intussusception. Zheng et al. in China were the first to report that the number of pediatric patients with intussusception decreased significantly in 2020 [14], as demonstrated by some recent studies [15,16,17,18]. Nevertheless, the in-depth influence of the COVID-19 pandemic on the diagnosis and treatment of intussusception for pediatric patients has not been well documented. 

Meanwhile, the effects of the COVID-19 pandemic on the operation of USGHER for intussusception have been largely unknown. In the present study, based on the data from one of the largest hospitals in Mainland China in a different period of COVID-19, we retrospectively analyzed and compared the clinical features and visiting processes of pediatric patients with intussusception who were treated by USGHER.

## 2. Materials and Methods

### 2.1. Patient Enrollment and Study Design

Patients younger than 14 years old who were diagnosed with intussusception and treated through USGHER from January to March in the respective years of 2019, 2020, and 2021 in our institution were enrolled in this study. Intussusception was defined as the invagination of one segment of intestine into a segment of distal intestine, and was diagnosed by experienced examiners with an abdominal ultrasound (US) [19]. Patients with suspected intussusception but without the US confirmation were excluded. Patients with intussusception receiving USGHER were evaluated by pediatric surgeons and the procedure was performed by both experienced pediatric surgeons and US specialists, according to the clinical features of intussusception and patients’ autonomous choices. 

Patients with intussusception receiving X-ray-guided pneumatic enema reduction, direct surgical treatment, and other non-invasive therapies were excluded. Patients combined with other acute abdominal diseases (such as appendicitis, pancreatitis, and intestinal obstruction of unknown origin) were excluded. Patients with missing demographic data were also excluded (Figure 1). As for recurrent intussusceptions after hospital discharge, we handle them as the same cases and analyzed the clinical features of their first visit.

According to the epidemic characteristics in Mainland China, 2019 was the first year before the outbreak of COVID-19, which is defined in the present study as the “pre-pandemic” period (Control Group); 2020 was the first year during the pandemic of COVID-19 and the most serious stage of epidemic situation, which is defined as the “pandemic” period (Study Group 1); 2021 was the second year during the pandemic of COVID-19 and the relatively stable stage of the epidemic situation, which is defined as the “post-pandemic” period (Study Group 2), although the COVID-19 pandemic is still ongoing. Our retrospective study was approved by the local Institutional Review Board, and written informed consent was obtained during the first emergency visit from all patients, which is consistent with the standards of the Declaration of Helsinki [20].

### 2.2. Data Collection and Outcome Measurement

The data of enrolled patients were retrospectively reviewed from electronic medical records and expectantly documented in the prepared tabulations, including demographic data (sex, age, address), epidemiological information (potential history of COVID-19 exposure or touch), symptoms and signs (vomiting, abdominal pain, bloody stool, palpable mass, fever, cough, nasal stuffiness, sore throat, headache, and so on), auxiliary investigations (abdominal US, chest computed tomography (CT), real-time polymerase chain reaction against SARS-CoV-2), features of USGHER (time of duration, pressure of enema, rate of success, complications), etc.

In order to more precisely assess how the COVID-19 pandemic would influence intussusception with USGHER for pediatric patients, we intentionally set up the time to intervention (TTI) for their visiting process in our emergency department (ED). To be specific, TTI 1 was defined as the time from the onset of symptoms associated with intussusception to the visit of the ED; TTI 2 from the visit to performing the abdominal US (i.e., the time of diagnosing intussusception in the ED); TTI 3 from the US to receiving USGHER; and TTI 4 as the length of stay in the ED.

The primary outcome of our study was the number of patients with intussusception receiving USGHER in the period of “pre-pandemic”, “pandemic”, and “post-pandemic” of COVID-19. The secondary outcome was the clinical features and TTIs among Control Group, Study Group 1, and Study Group 2.

### 2.3. Statistical Analysis

Data of the present study were entered into the database by one author and checked by another. Quantitative variables were reported as mean with standard deviation (SD) or median somewhere and compared using the analysis of variance (ANOVA) between the above three groups. Categorical variables were presented as numbers with their frequencies as proportions (%) and compared using the Pearson’s chi-squared test and Fisher’s exact test. All statistical analyses were carried out using IBM SPSS 25.0 statistical software (SPSS Inc., Chicago, IL, USA). Difference with a two-sided *p* value less than 0.05 was considered statistically significant.

## 3. Results

In the present study, we ultimately enrolled 246 eligible patients, including 90 in Control Group, 70 in Study Group 1, and 86 in Study Group 2 (Table 1). The total cases in Study Group 1 decreased significantly from those in Control Group and Study Group 2 (*p* = 0.042), while those in Study Group 2 were mostly similar to those in Control Group (*p* = 0.215). All groups had a male preference (*p* = 0.125). Patients’ age and their prior history of intussusception in each group present no differences (*p* = 0.313, *p* = 0.114, respectively). 

The number of patients addressed out of Chengdu in Study Group 1 was statistically smaller than that of Control Group and Study Group 2 (*p* = 0.027), as well as the history of treatment in other hospitals (*p* = 0.013), while those between Control Group and Study Group 2 present no significant differences (*p* = 0.112, *p* = 0.323, respectively). As for the clinical symptoms or signs associated with intussusception, the proportion of vomiting, abdominal pain, palpable mass, and fever in each group showed no notable differences (*p* = 0.374, *p* = 0.153, *p* = 0.318, *p* = 0.936, respectively), while that of bloody stool in Study Group 1 increased significantly than that in other two groups (*p* = 0.007).

According to the findings of a specialized US examination, both solitary lesion and ileocolic type for intussusception in each group accounted for the majority (*p* = 0.715, *p* = 0.357, respectively). There were no significant differences among the three groups in terms of the transverse diameter at the widest point of intussusception and swollen lymph nodes (*p* = 0.354, *p* = 0.147, respectively). The longest invaginated length of intussusception in Study Group 1 was statistically longer than that of Control Group and Study Group 2 (*p* = 0.042), while that between Control Group and Study Group 2 had no significant difference (*p* = 0.516).

As for the features of USGHER, comparisons of neither pressure nor the duration of enema reduction present statistical significances among these three groups (*p* = 0.091, *p* = 0.085, respectively), although those in Study Group 1 were slightly higher or longer than those in Control Group and Study Group 2. Without any case of perforation, the rate of success to reduction in each group was over 95%, while the rate of recurrence was lower than 4%, with no notable differences (NA, *p* = 0.823, NA, respectively). The expense in the ED in Study Group 1 was also slightly higher than that of Control Group and Study Group 2, while their comparison did not present any statistical difference (*p* = 0.072).

In terms of the TTIs, as we previously defined, both TTI 1 and TTI 2 in Study Group 1 were significantly longer than that of Control Group and Study Group 2 (*p* = 0.036, *p* = 0.031, respectively), while the TTIs in Control Group and Study Group 2 were much closer to one another (*p* = 0.125, *p* = 0.368, respectively). Furthermore, comparisons of neither TTI 3 nor TTI 4 among each group present significant differences (*p* = 0.334, *p* = 0.518, respectively). According to whether his/her temperature was higher than 37.3 °C, we subsequently divided patients from 2020 and 2021 into a fever group and a non-fever group, and compared their TTIs (Table 2). We found that comparisons of TTI 1, TTI 2, and TTI 4 between fever group and non-fever group were statistically different (*p* = 0.034, *p* = 0.028, *p* = 0.041, respectively), while that of TTI 3 was not (*p* = 0.256).

During the pandemic of COVID-19, we gradually took some related screening measures for pediatric patients with intussusception, with an improvement in the worldwide understanding of this communicable disease (Table 3). In 2020, we identified 21.4% patients with fever and 17.1% with non-gastrointestinal symptoms, compared to 20.9% and 16.3% in 2021. All patients were required to check the history of his/her COVID-19 epidemiology. There were 28.6%, 15.7%, and 10% of patients who underwent a CT chest scan, antibody detections, and nucleic acid detections SARS-CoV-2 in 2020, while there were 17.4%, 6.9%, and 26.7% in 2021, respectively. There were no patients with a history of COVID-19 exposure or touch and no COVID-19-confirmed patients in our study.

## 4. Discussion

During the pandemic of COVID-19, we gradually took some epidemic-preventative measures for patients visiting our institution according to the national epidemic policies and the hospital’s infectious disease strategies, such as setting up dedicated entrances and exits for staff and patients, checking health codes and trip codes, inquiring about the epidemiological history of patients and families, asking about respiratory and non-respiratory symptoms, treating fever and non-fever patients in different reception rooms, performing imaging examinations, testing virus antibodies or nucleic acid, etc. According to our analyses (Table 3), there was more CT chest scans (28.6% vs. 17.4%) and detections of antibodies against SARS-CoV-2 (15.7% vs. 6.9%) in 2020 but less detections of nucleic acid (10% vs. 26.7%), compared with those in 2021, indicating the involvement and development of our ability to diagnose COVID-19 in the “pandemic” and “post-pandemic” periods of COVID-19. Theoretically, those complicated but mandatory procedures might hinder emergency medicine.

Since the outbreak of COVID-19, this epidemic has brought about widespread restrictions on socializing, travel, business, education, and healthcare. The compulsory strategies to prevent the spread of COVID-19 by both countries and hospitals, as well as the fear of exposure and risk of infection for patients themselves [4], has changed the epidemiology of various diseases, such as bronchiolitis and asthma [5], seasonal influenza [6], acute appendicitis [7], etc. As for intussusception for pediatric patients, Zheng et al. in China were the first to report that the COVID-19 pandemic and its resultant quarantine strategies have significantly reduced the incidence of intussusception [14]. 

Some later studies also documented that the COVID-19 pandemic and the implementations of preventative measures to combat this communicable disease have notably influenced the epidemiology of intussusception [15,16,17,18]. Meanwhile, Handa et al. demonstrated that social distancing has resulted in a significant decline in intussusception [18]. Our study also noted the epidemiological feature of intussusception during the COVID-19 pandemic, in which the number of patients receiving USGHER in 2020 decreased significantly from that in 2019 (*p* = 0.042). At the same time, we, for the first time, reported that cases of intussusception receiving USGHER in 2021 were mostly close to those in 2019 (*p* = 0.215), indicating that the epidemiological influences of COVID-19 on intussusception may have weakened.

Besides the above impacts on incidence, the COVID-19 pandemic might lead to a diagnostic delay of intussusception for pediatric patients. In our present study, the number of patients receiving USGHER from outside Chengdu decreased significantly in 2020 (*p* = 0.027), while the visiting history of treatment in other hospitals increased notably (*p* = 0.013). These epidemiological phenomena echoed the significantly prolonged TTI 1 (i.e., the time from the onset of symptoms associated with intussusception to the visit of the ED) during the period of the COVID-19 “pandemic” (*p* = 0.036), which might be related to the fear of exposure and risk of infection for patient visits in large hospitals [4]. 

Meanwhile, we found that TTI 2 (i.e., the time from the visit of the ED to diagnosing intussusception) during the COVID-19 “pandemic” period was also significantly longer (*p* = 0.031), which might be attributed to the complicated but compulsory preventative measures to combat COVID-19 at the beginning of this epidemic. Park et al. reported that the time to diagnose intussusception was slightly longer during the “pandemic” period compared with the “pre-pandemic” period (*p* = 0.063) [16], while Lee et al. reported that the time to intervention from visiting the ED to performing the US presented no notable difference (*p* = 0.575) [17].

The COVID-19 pandemic might also lead to the deterioration of clinical manifestations for pediatric patients with intussusception. Given what we know, bloody stool, i.e., “currant jelly stool”, is one of the main clinical manifestations of aggravations of patients’ health condition for intussusception [8,9,10]. As we demonstrated in the present study, the number of patients with bloody stool during the period of COVID-19 “pandemic” period was statistically more than that during the period of “pre-pandemic” and “post-pandemic” (*p* = 0.007). Through the records of US examinations, we also correspondingly found that the longest invaginated length of intussusception during the COVID-19 “pandemic” period was statistically longer than that of the other two groups (*p* = 0.042), indicating the possible cause of bloody stool. 

Our demonstration is in agreement with what Park et al. reported, in which the proportion of a more serious condition for pediatric patients with intussusception was higher after the COVID-19 outbreak (*p* = 0.033) [16]. However, Zheng et al. mentioned that the patients’ symptoms with intussusception in 2020 did not become more severe than those in 2018 and 2019, because there were no significant differences regarding the number of cases of recurrence and those requiring emergency surgical intervention, as well as the length and diameter of the involved intestine (*p* > 0.05) [14].

As one of the most effective invasive treatments of intussusception for pediatric patients, USGHER has been increasingly performed in more and more medical institutes [11,12,13]. Hereby, we, for the first time, revealed the exact impacts of the COVID-19 pandemic on the operation of USGHER. According to our analysis, the pressure and duration of enema reduction during the COVID-19 “pandemic” period seemed to be slightly higher or longer than those in “pre-pandemic” and “post-pandemic” periods, while their comparisons present no statistical differences (*p* = 0.091, *p* = 0.085; respectively). Meanwhile, patients in a different period of COVID-19 had a very close and high success rate of reduction (*p* = 0.823), which was similar to the reported data in previous literature [11]. 

For pediatric patients with intussusception in these three groups, there was no perforation case involved, and recurrence of intussusception occurred in few cases. What is more, the comparisons referring to expense in the ED or TTI 4 (i.e., length of stay in the ED) were not significant (*p* = 0.072, *p* = 0.518; respectively), which basically corresponds to what Park et al. [16] and Lee et al. [17] reported. Finally, we also found that comparison of TTI 3 (i.e., the time from diagnosing intussusception to receiving USGHER) among each period was not statistically significant (*p* = 0.334), revealing the timely operation of USGHER for those patients without obvious influences by the COVID-19 pandemic. Therefore, our demonstrations indicate that although the COVID-19 pandemic might have caused a delay in diagnosing intussusception and the exacerbation of patients’ symptoms, it did not significantly affect the operation of USGHER and patients’ clinical outcome.

Our study has several limitations due to its retrospective nature. Firstly, our analysis was based on data collected from one single affiliated hospital, which might reduce the statistical power between factors and outcomes of intussusception for pediatric patients. Secondly, there were no cases with perforation and few cases with recurrent intussusception for patients undergoing USGHER, which were not comparable among the “pre-pandemic”, “pandemic”, and “post-pandemic” periods of COVID-19. Furthermore, our study lacked follow-up information, which reduces the data that can be used to further evaluate the long-term impact of COVID-19 on intussusception. Finally, intussusception has been widely reported as the initial symptom of pediatric patients in association with COVID-19 [21,22,23,24] and the role of SARS-CoV-2-mediated intestinal lymphoid hypertrophy in the etiopathogenesis of intussusception has been recently documented [25], while our study lacked COVID-19-confirmed cases and direct histological evidences. Therefore, prospective, large-scale, multi-center studies that include follow-up are required to overcome these limitations.

## 5. Conclusions

In a word, the number of pediatric patients with intussusception receiving USGHER decreased significantly during the initial period of COVID-19. Although the implementation of preventative measures to combat the COVID-19 pandemic might delay the timely diagnosis of intussusception and worsen the clinical symptoms for pediatric patients, it did not significantly affect the operation of USGHER and patients’ clinical outcome.

## Figures and Tables

**Figure 1 jcm-11-04473-f001:**
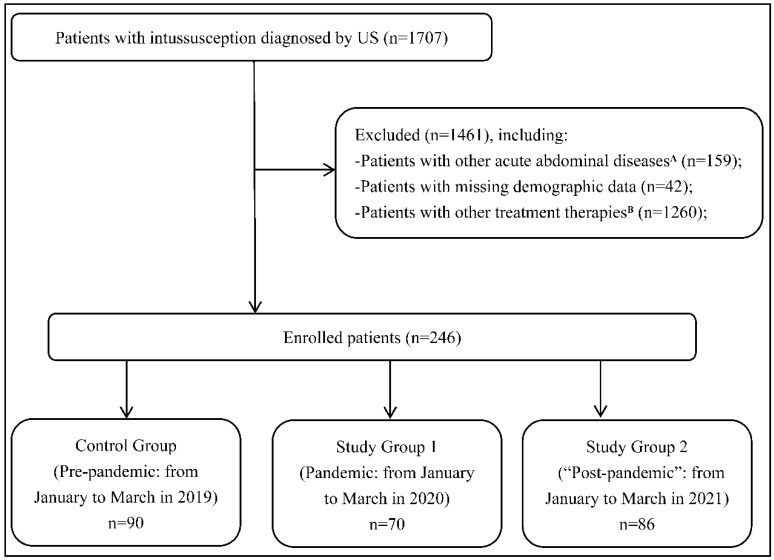
Flow chart of the present study. There were 1707 patients who were diagnosed with intussusception using US. We excluded 1461 patients who did not meet the inclusive criteria and ultimately enrolled 246 ones in this study. There were 90 patients in Control Group (pre-pandemic), 70 patients in Study Group 1 (pandemic), and 86 patients in Study Group 2 (“post-pandemic”). ^A^: Refers to patients with intussusception combined with other acute abdominal diseases, such as appendicitis, pancreatitis, and intestinal obstruction of unknown origin. ^B^: Refers to patients with intussusception receiving X-ray-guided pneumatic enema reduction, direct surgical treatment, and other non-invasive therapies. Note: US = ultrasound.

**Table 1 jcm-11-04473-t001:** Clinical characteristics of pediatric patients with intussusception in different periods of the COVID-19 pandemic.

Factor	Control Group	Study Group 1	Study Group 2	*p* ^A^
Total cases	90	70	86	0.042
Sex, male	60(66.7%)	45(64.3%)	55(63.9%)	0.125
Age, mons.				0.313
Mean	21.12 ± 11.24	22.35 ± 12.46	22.15 ± 14.26	
Median (range)	22.1(2.1–60.5)	23.1(1.2–54.3)	23.1(1.7–61.2)	
Address, out of Chengdu	52(57.8%)	30(42.8%)	49(56.9%)	0.027
Past history of intussusception	9(10.0%)	6(8.5%)	8(9.3%)	0.114
History of treatment in other hospitals	25(27.7%)	27(38.6%)	26(30.2%)	0.013
Symptoms or signs				
Vomiting	55(61.1%)	41(58.6%)	52(60.5%)	0.374
Abdominal pain	58(64.4%)	44(62.8%)	55(63.9%)	0.153
Bloody stool	20(22.2%)	26(37.1%)	21(24.4%)	0.007
Palpable mass	42(46.6%)	34(48.5%)	40(46.5%)	0.318
Fever (T ≥ 37.3 °C)	19(21.1%)	15(21.4%)	18(20.9%)	0.936
US for intussusception				
Number, solitary	85(94.4%)	66(94.3%)	80(93.0%)	0.715
Type, ileocolic	79(87.8%)	62(88.6%)	77(89.5%)	0.357
Length ^B^, cm.				0.042
Mean	4.1 ± 1.2	5.8 ± 1.3	3.9 ± 0.5	
Median (range)	3.2(2.5–7.5)	4.1(3.3–9.0)	3.1(2.6–6.5)	
Width ^C^, cm.				0.354
Mean	3.6 ± 0.5	3.3 ± 0.6	3.6 ± 0.4	
Median (range)	3.4(2.4–4.5)	3.5(2.5–4.6)	3.3(2.5–4.3)	
Swollen lymph nodes ^D^	25(27.7%)	20(28.6%)	26(30.2%)	0.147
USGHER				
Pressure, cmH_2_O				0.091
Mean	102.1 ± 22.3	108.6 ± 24.2	103.8 ± 21.5	
Median (range)	100(80–120)	100 (80–140)	100 (80–140)	
Time of duration, min.				0.085
Mean	4.3 ± 0.8	4.9 ± 1.5	4.4 ± 1.1	
Median (range)	4.1(1.5–15.2)	4.4(2.5–20.4)	4.1(2.1–18.5)	
Success to reduction	87(96.7%)	67(95.7%)	83(96.5%)	0.823
Perforation	0	0	0	NA
Recurrence	3(3.3%)	2(2.9%)	3(3.5%)	NA
Expense in ED, RMB				0.072
Mean	4355.2 ± 357.1	4863.1 ± 331.4	4536.3 ± 401.7	
Median (range)	4255(4112.5–5237.6)	4530(4168.2–5486.9)	4436(4143.5–5343.8)	
Time span ^E^, hours				
TTI 1	15.2 ± 6.1	22.5 ± 4.5	14.5 ± 7.8	0.036
TTI 2	1.1 ± 0.4	2.2 ± 0.9	1.3 ± 0.7	0.031
TTI 3	0.6 ± 0.2	0.8 ± 0.4	0.7 ± 0.4	0.334
TTI 4	2.5 ± 0.7	3.1 ± 1.1	2.7 ± 1.4	0.518

^A^: Refers to the comparison among these three groups; ^B^: Refers to the longest invaginated length of intussusception; ^C^: Refers to the transverse diameter at the widest point of intussusception; ^D^: Refers to the regional lymph nodes measuring over 1 cm in diameter on the mesentery; ^E^: TTI 1 is defined as the time from the onset of symptoms associated with intussusception to the visit of ED; TTI 2 from the visit to performing the abdominal US (i.e., the time of diagnosing intussusception in ED); TTI 3 from the US to receiving USGHER; TTI 4 as the length of stay in ED. Note: COVID-19 = coronavirus disease 2019; T = temperature; US = ultrasound; USGHER= ultrasound-guided hydrostatic enema reduction; ED = emergency department; NA = not applicable; TTI = time to intervention.

**Table 2 jcm-11-04473-t002:** Comparisons of TTIs between fever group and non-fever group for pediatric patients with intussusception during the COVID-19 pandemic (in 2020 and 2021).

Time span	Fever Group (n = 33)	Non-fever Group (n = 123)	*p*
TTI 1, hours	12.5 ± 3.2	20.1 ± 5.4	0.034
TTI 2, hours	2.3 ± 1.2	1.4 ± 0.5	0.028
TTI 3, hours	0.9 ± 0.4	0.8 ± 0.2	0.256
TTI 4, hours	4.2 ± 1.2	3.1 ± 0.8	0.041

Note: TTI = time to intervention; COVID-19 = coronavirus disease 2019.

**Table 3 jcm-11-04473-t003:** Related screening measures of COVID-19 for pediatric patients with intussusception.

Factor	Study Group 1 (n = 70)	Study Group 2 (n = 86)
Fever (T ≥ 37.3 °C)	15 (21.4%)	18(20.9%)
Non-gastrointestinal Symptoms ^A^	12 (17.1%)	14(16.3%)
Inquiring the COVID-19 epidemiological history	100	100
Chest CT scan	20 (28.6%)	15(17.4%)
Detection of antibodies against SARS-CoV-2	11(15.7%)	6(6.9%)
Detection of nucleic acid against SARS-CoV-2	7 (10%)	23 (26.7%)
History of COVID-19 exposure or touch	0(0%)	0(0%)
COVID-19 confirmed	0(0%)	0(0%)

^A^: Refers to cough, sputum, rhinorrhea, nasal stuffiness, myalgia, sore throat, headache, anosmia, dyspnea, pneumonia, and so on. Note: COVID-19 = coronavirus disease 2019; T = temperature; CT = computerized tomography.

## Data Availability

The datasets analyzed during the current study are not publicly available right now because these materials also form part of an ongoing study, but are available from the corresponding author on reasonable request.

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
