# Peer review of "How Does the COVID-19 Pandemic Affect Pediatric Patients with Intussusception Treated by Ultrasound-Guided Hydrostatic Enema Reduction?"

_jcm, 2022, doi:10.3390/jcm11154473_

Round 1

Reviewer 1 Report

The study evaluates the management of intussusception in the preceding, during and post-pandemic period. The result is quite obvious, however with modifications it can be accepted.

Line 74: remove: As far as we know, our study was the first attempt to do such a work.

Line 79: It is not easy to understand: from January to March in 2019, 2020 and 2021 in our institution

were enrolled in this study

Line 83: It is not easy to understand : Patients with intussusception receiving USGHER were evaluated by pediatric surgeons and performed by both experienced pediatric surgeons and US specialists, according to the clinical features of intussusception and patients’ autonomous choices

From 206 to 235 reduce this generic part.

Reviewer 2 Report

Authors showed the relation between the influence of the spread of COVID-19 infection and pediatric patients with intussusception restricted to the patients treated by USGHER. They divided observation periods into 3; pre, post and pandemic. Because the epidemic of infectious disease is different in each region and the number of COVID-19 infected patients fluctuate in several month cycle, authors should show “pandemic” and “post pandemic” using the objective data such as the number of patients in each period.

No objective data relation between patients and COVID- 19 infection were shown in the manuscript. The patients with intussusception analyzed in this manuscript were restricted to the children who treated by USGHER. Moreover, there no relation between COVID-19 infection and USGHER. Many patients many patients who treated by other methods were excluded. Epidemiological analysis in this paper may not elucidate the relation between the COVID-19 epidemic and intussusception in children because of the large bias.

Reviewer 3 Report

Congratulations on your excellent work.

It is worthy for presenting what we feel in the medical field  with scientific evidence. I totally agree with your results. 

This paper is meaningful to report the changes of the incidence of intussusception during COVID-19 pandemic period. 

Also, you showed excellent treatment outcomes. I think it is result of all medical staffs that overcome the terrible medical environment. 

I have two questions in your results. 

1) Why did you compare two groups of fever and non-fever group?

   The development of fever reflects the severity of intussusception and it is one of indication of surgical treatment. Could you add your interpretation of these results more in detail?

2) Could you explain the meaning of Table 3?

   I'm afraid to make confusion to interpret the data about detection of antibodies/nucleic acid against SARS-CoV-2 between COVID-19-confirmation. I need clear description about these result, and please add clear interpretation in discussion part.    

Round 2

Reviewer 1 Report

From 206 to 235 reduce this generic part.

Reviewer 2 Report

The reviced manuscript might be acceptable for the journal.
